



# Improved Extended-Range Prediction of Persistent Stratospheric Perturbations using Machine Learning

Raphaël de Fondeville[1], Zheng Wu[2], Enikő Székely[1], Guillaume Obozinski[1], and Daniela I.V. Domeisen[3,2]

[1]Swiss Data Science Center, ETH Zurich and EPFL, Lausanne, Switzerland
[2]ETH Zurich, Zurich, Switzerland
[3]Université de Lausanne, Lausanne, Switzerland

**Correspondence:** Raphaël de Fondeville (raphael.de-fondeville@epfl.ch)

**Abstract.** On average every two years, the stratospheric polar vortex exhibits extreme perturbations known as Sudden Stratospheric Warmings (SSWs). The impact of these events is not limited to the stratosphere, but they can also influence the weather at the surface of the Earth for up to three months after their occurrence. This downward effect is observed in particular for SSW events with extended recovery timescales. This long-lasting stratospheric impact on surface weather can be leveraged

to significantly improve the performance of weather forecasts on timescales of weeks to months. In this paper, we present a fully data-driven procedure to improve the performance of long-range forecasts of the stratosphere around SSW events with an extended recovery. We first use unsupervised machine learning algorithms to capture the spatio-temporal dynamics of SSWs and to create a continuous scale index measuring both the frequency and the strength of persistent stratospheric perturbations. We then uncover three-dimensional spatial patterns maximizing the correlation with positive index values allowing us to assess

when and where statistically significant early signals of SSW occurrence can be found. Finally, we propose two machine learning (ML) forecasting models as competitors for the state-of-the-art sub-seasonal numerical prediction model ECMWF S2S: while the numerical model performs better for lead times up to 25 days, the ML models offer better predictive performance for greater lead times. We leverage our best performing ML forecasting model to successfully post-process numerical ensemble forecasts and increase their performance by up to 20%.

## 1 Introduction

In both hemispheres and during winter, the atmosphere above the polar regions is characterized by eastward winds centered around the poles with a mid-winter peak in wind intensity, the so-called "polar vortex". On average once every two years, in the Northern Hemisphere, upward propagating Rossby waves can disturb the polar vortex and induce a sudden warming of the polar stratosphere. Known as Sudden Stratospheric Warmings (SSWs; Baldwin et al., 2021), these events impact not only

the stratosphere but they also strongly influence the weather at the Earth's surface up to three months after their occurrence (Baldwin and Dunkerton, 2001). Therefore, SSW events are considered an important source of predictability of surface weather on sub-seasonal timescales ranging from 2 weeks to 2 months. Improving the prediction of SSW events may therefore help enhance the forecast performance of surface weather (Sigmond et al., 2013; Domeisen et al., 2020a).





The dynamics behind SSWs are not yet fully understood (Baldwin et al., 2021) and hundreds of contributions have been made
on the topic, including for instance a classification of stratospheric perturbations based on their influence on the troposphere
to uncover common dynamical precursors (Runde et al., 2016), or the quantification of predictive skill increase after SSWs
occurrences (Karpechko et al., 2017a; Karpechko, 2018; Domeisen et al., 2020b; Wu et al., 2022). The predictive skill of the
onset of SSWs by state-of-the-art numerical prediction models remains limited to about 1-2 weeks (Domeisen et al., 2020a).
Improving the predictability of SSWs would significantly contribute to better sub-seasonal to seasonal weather forecasts on
timescales of several weeks to months.

SSWs with a stronger and more persistent impact on the stratosphere are generally associated with slowly descending zonal
wind anomalies (Kodera et al., 2000) within the stratosphere, a phenomenon also known as Polar Night Jet Oscillation events
(PJO) (Kuroda and Kodera, 2001, 2004). In these cases, the vortex is not only subject to larger-scale perturbations, but the
recovery of the vortex takes a longer amount of time, up to 2 to 3 months, to retrieve its quasi-circular shape above the pole,
and the disturbances tend to reach further down, all the way to the lower stratosphere, where they have a higher potential for
affecting the underlying troposphere. Hence, SSWs with slow recovery are likely to influence the weather at the Earth's surface
for a prolonged period of time, thereby offering an opportunity to increase weather predictability; this work is thus focused on
this particular kind of persistent event.

One difficulty in uncovering relevant precursors to SSW events is the need to analyze large quantities of three-dimensional
spatio-temporal data of different physical variables with complex interactions. For this reason, existing studies have relied on
domain expertise to focus on regions of the atmosphere and mechanisms assumed relevant for SSWs. From the data analysis
point of view, conducting a systematic search of the atmospheric dynamics to localize and uncover statistically significant sig-
nals with predictive potential for SSWs with long-lasting impacts would help getting a deeper understanding of their dynamics.
Machine Learning (ML) and, more generally, data science techniques offer fully data-driven alternatives for a systematic search
of new insights about relevant atmospheric dynamics that can be applied to large quantities of data. ML approaches can also
represent competitive alternatives to numerical forecasts that can be leveraged to improve weather prediction performance.

Indeed, data-driven techniques have already been successfully applied to improve the understanding of SSWs dynamics.
For instance, an application of Empirical Orthogonal Functions (EOFs) on potential vorticity data during SSW events revealed
unknown drivers as well as early signals of SSW onset (Rongcai and Cai, 2006; Lu and Ding, 2015; Lu et al., 2016). The vortex
geometry, and its evolution, has been analyzed in detail through a combination of moment analysis and extreme value theory
(Mitchell et al., 2011) and through methodologies from computer vision (Lawrence and Manney, 2018); the latter allows
for a detection and representation of the vortex evolution in three dimensions providing a tool to visualize the vortex pre-
conditioning. The relationship between SSWs and planetary wave activity was explored using composite methods (Bancalá
et al., 2012), while the influence of different states of the stratosphere on SSW occurrence has been assessed through the
computation of conditional probabilities (Jucker and Reichler, 2018). Furthermore, a wide range of studies have focused on
classifying SSW events into multiple categories, such as different patterns of planetary wave activity (Bancalá et al., 2012; Wu
et al., 2021), event intensity (Blume et al., 2012), data-driven categories (Coughlin and Gray, 2009; Lawrence and Manney,
2020), or the geometry of the vortex perturbation, i.e., split or displacement (Hannachi et al., 2011).





Following the suggestion of Cohen et al. (2019), we go beyond exploration and classification and propose an fully data-driven ML procedure that can be leveraged to improve long-range numerical surface weather forecasts. Our methodology can be divided into three steps: first, we apply a classical dynamics-driven unsupervised ML algorithm to define a continuous scale index quantifying the strength and occurrence of SSWs with slow recovery. Secondly, the index is used as a predictive target to find spatio-temporal atmospheric patterns with statistically significant predictive power. Finally, supervised learning is used to predict the index from the atmospheric regions and forecast lead time found in the previous step. In other words, we first answer the question of what quantity to forecast, then of where and when can we find relevant predictors, and we finally determine the lead time at which we can produce a skillful forecast. The best performing ML algorithm found in the last step of the procedure is used to post-process numerical forecasts and to obtain an increase of up to $20\%$ in performance. Our methodology is similar to the work of Kretschmer et al. (2017), where a causal discovery algorithm is used to find relevant predictors, however it differs in two main ways. First, we devise a procedure allowing us to analyze a much larger quantity of data, since causal methods usually suffer from computational limitations. Secondly, while Kretschmer et al. (2017) only investigate the skill horizon of ML-based forecasts, we additionally integrate our findings to improve the performance of standard numerical forecasts. ML algorithms have also been applied in similar contexts (Blume and Matthes, 2012; Minokhin et al., 2017), however in these works relevant predictors had to be carefully designed ahead of time using knowledge of the physical process, while in our approach relevant three-dimensional patterns are directly learned from the original data.

The paper is structured as follows: Section 2 summarizes the different data sources and their characteristics. In Section 3, we propose a continuous scale index based on the vertical variation of the stratospheric temperature anomalies characterizing SSW with a slow recovery, i.e., with a long-lasting impact. Section 4 leverages supervised learning to detect spatio-temporal stratospheric patterns with statistically significant predictive power. In Section 5, we propose several ML models to forecast the index proposed in Section 3 and assess their performance. We then use the best model to post-process the forecasts from a numerical model and quantify the performance improvement. Finally, Section 6 briefly summarizes our contribution and gives potential directions for improvements.

## 2 Data

The analysis presented in this paper relies on the ERA-Interim reanalysis data produced by the European Center for Medium-Range Prediction (ECMWF) (Dee et al., 2011), i.e., climate model output with assimilated observational data every 6 hours from January 1st, 1979 to August 31st, 2019, simulated on a grid with a resolution of $0.75°$ and 60 pressure levels. Multiple atmospheric parameters are available, but to monitor sudden stratospheric warmings, we focus on temperature and potential vorticity.

To characterize stratospheric perturbations and identify their precursors, we need to analyze a large quantity of reanalysis data: for each physical quantity, the ERA-Interim provides $N \approx 59000$ temporal data points at every grid cell and pressure level. To reduce the quantity of the data to be analyzed, except when stated otherwise, we focus on the Northern Hemisphere extended winter, i.e., October to April, reducing the number of data points to $N = 33960$. Even though SSWs are also found





in the Southern Hemisphere, the Northern Hemisphere events are much more frequent; we therefore limit the analysis to this region of the globe. With 15 pressure levels in the troposphere, from 800hPa to 225hPa, and 14 pressure levels from 200hPa to 1hPa, including only data for the Northern Hemisphere at a $0.75°$ resolution yields a grid with 1.6 million cells per physical

quantity at each time step. We thus need to further reduce the dimensionality of the data to enable tractable and efficient data analysis.

For temperature, we proceed as in Blume et al. (2012) and Hitchcock et al. (2013a) by computing the polar cap average of temperature anomalies using every grid cell poleward of $60°$ N. As the goal is to analyze temperature variations inside the vortex, we focus on 12 stratospheric pressure levels, i.e., $150, 125, 100, 70, 50, 30, 20, 10, 7, 5, 2, 1$hPa, with lower and higher

levels corresponding to the "top" and "bottom" of the vortex, respectively.

Alternatively, to directly study the perturbations of the vortex, and not only the vertical structure of temperature anomalies, we analyze potential vorticity (PV), a quantity used in atmospheric dynamics and meteorology to characterize rotational fluids with a vertical stratification, where it is conserved for adiabatic friction-less motion. Because the vortex can be displaced to fairly low latitudes, we analyze a slightly larger region for PV as compared to temperature, i.e., all grid points poleward of

$30°$ N. Additionally, since we are interested not only in the structure of the vortex in the stratosphere but also in potential tropospheric precursors, we focus on 29 pressure levels, from 800hPa to 1hPa, with $0.75°$ spacing, i.e., $d = 480 \times 81 = 38880$ grid cells, yielding a vector of size 1.1 million per temporal snapshot. While such a dimensionality could reasonably be handled by ML algorithms, when trying to analyze the dynamics of the stratosphere, see Section 4, by studying simultaneously multiple dates and times over periods of weeks to months, the dimensionality of the data needs to be further reduced. We follow

the approach described in DelSole and Tippett (2015) to obtain an orthogonal functional basis over the spherical cap, i.e., a counterpart of the Fourier basis over a sub-region of a sphere capturing the characteristics of the data at different spatial scales, while taking into account the rotational invariance of the cap. Then, for each individual time step, the gridded data is projected onto the basis, yielding one coefficient per basis element. The new data representation is then obtained from the vector of basis coefficients, which we choose to have a lower dimension than the original grid size while preserving the field structure

as accurately as possible. Figure 1 gives an example of the PV field reconstructed using only the first 150, 300 and 600 lower spatial-frequency components of the functional basis. Using a few hundreds coefficients, as illustrated in Figure 1, already yields an accurate representation of the large-scale features in the reconstructed fields. In Section 4, we will therefore use a limited number of basis coefficients to represent the gridded set on each pressure level, which enables us to simultaneously analyze data at multiple levels and time steps.

Finally, we aim to illustrate how machine learning techniques can be used to improve the long-range performance of current numerical forecasts. To do so, we leverage one dataset of forecasts from the subseasonal to seasonal (S2S) prediction database (Vitart et al., 2017): in this project, the numerical forecasts are produced by running weather models initialized with the state of the climate at the time of initialization. The ECMWF prediction system, which we analyze in this work, provides a set of hindcasts, i.e., forecasts produced a-posteriori for validation purposes, initialized every two to three days, and using the ERA-

Interim reanalysis as initialization. To account for uncertainty in the initial conditions and hence the subsequent dynamical trajectory, not only one but multiple hindcasts are produced from slightly perturbed initial conditions to form a set of equally




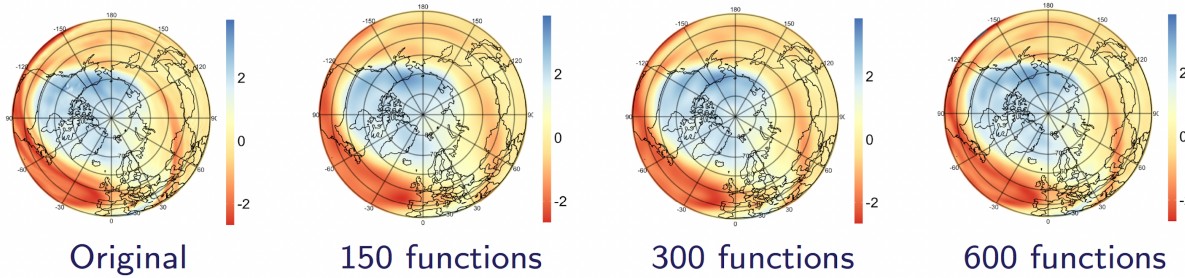

**Figure 1.** Original (left) and reconstructed potential vorticity field using an increasing number (150, 300 and 600) of the functional basis. The orthogonal functional basis over the spherical cap was computed using the approach in DelSole and Tippett (2015).

likely forecasts called an ensemble. In particular, the ECMWF model from the S2S database provides hindcasts consisting of 11 members for lead times up to 47 days after the initialization date. These forecasts are available for daily mean values at the pressure levels of 100, 50 and 10hPa and at a horizontal resolution of 2.5° latitude and longitude. Processing the lower resolution gridded data with a limited number of pressure levels and with the help of a lower resolution functional basis yields low-dimensional data representations similar to those obtained in Section 3 and 4. In this work, the ECMWF hindcasts will serve as a baseline for performance comparison in Section 5 and will be post-processed using machine learning forecasts to attempt improving their performance

## 3   Unsupervised characterization of SSW dynamics

Most classical characterizations of sudden stratospheric warmings are binary indices that usually require a pre-defined threshold, e.g., most commonly, the inversion of the zonal mean zonal wind at 60° latitude and 10 hPa, combined with a temporal "separation" rule for consecutive events (Charlton and Polvani, 2007). The choice of SSW definition can yield results that are not necessarily consistent between studies (Butler and Gerber, 2018); see Butler et al. (2015), Butler et al. (2017) and Baldwin et al. (2021) for thorough reviews of existing definitions. Using data-driven techniques on carefully chosen physical quantities, Coughlin and Gray (2009) conclude that characterization through a strict classification should be relaxed in favor of a "continuum" of perturbations to better reflect the complexity of the underlying atmospheric dynamics.

From a machine learning perspective, the limited number of events in the database and the uncertainty of the event start dates make the application of most algorithms using binary indices difficult and potentially inconsistent. For these reasons, we first define a continuous scale index to quantify the strength and the occurrence of the vortex perturbations. For this task, we use simple unsupervised machine learning algorithms, i.e., algorithms that aim at discovering dynamical patterns in the data without an auxiliary source of information, including potentially unknown patterns.

We start by analyzing stratospheric perturbations as captured by temperature anomalies using an analysis similar to Blume et al. (2012) and Hitchcock et al. (2013a): we apply Principal Component Analysis (PCA) on polar cap averaged temperature anomalies at 12 pressure levels between 150hPa and 1hPa. PCA uncovers patterns, also called principal directions, or PC





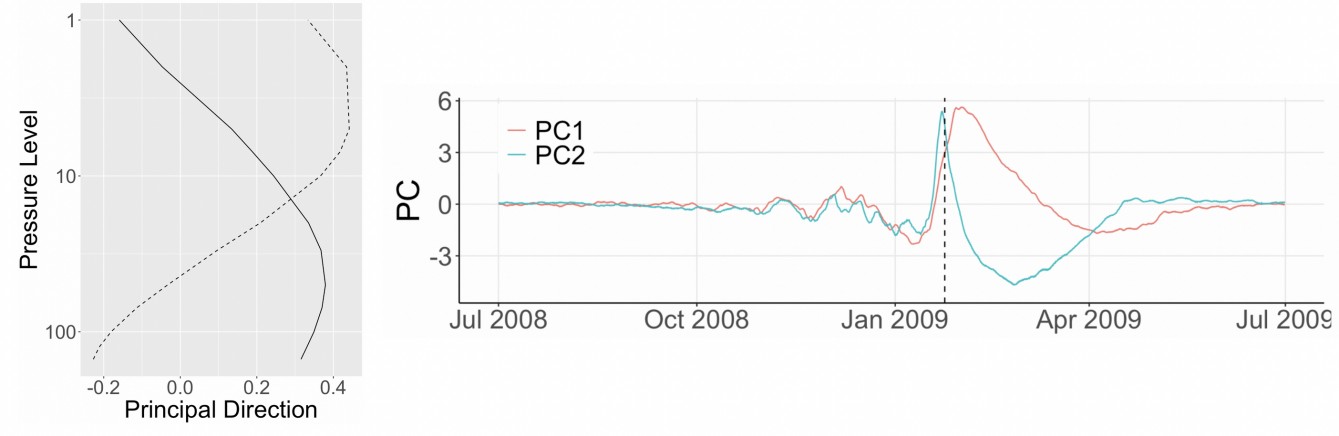

**Figure 2.** Left: First (solid) and second (dashed) principal direction of maximum variance obtained from a PCA on polar cap averaged temperature anomalies at pressure levels between 150 and 1hPa. Right: First (red) and second (green) Principal Components ($PC$) for winter 2008/09 obtained using PCA on polar cap averaged temperature anomalies at levels $150, 125, 100, 70, 50, 30, 20, 10, 7, 5, 2,$ and 1hPa. Vertical dashed line: date of first wind reversal at $60°$ and 10hPa.

directions, that provide data projections whose variance is maximized. Thus, the first PC direction provides the temperature profile showing the largest variation in amplitude, while the second PC direction represents the largest perturbation pattern once the first PC direction has been removed from the data, and so on. From a practical point of view, we apply PCA on vectors of size $d = 12$, each component corresponding to the spherical cap average of temperature anomalies at each pressure level. We note that such an approach focuses on the variability of temperature patterns observed at individual time steps and does not

account for dynamical properties of the phenomenon studied.

Figure 2 shows the vertical distribution of the first two PC directions: results are similar to Hitchcock et al. (2013a) as we observe that none of the patterns has exclusively positive (or negative) values suggesting that temperature perturbations do not occur simultaneously along the vortex vertical structure. Indeed, the vertical structure of the dominant modes indicates that the stratosphere can be separated into two layers whose dynamics during sudden stratospheric warmings is likely to differ.

Figure 2 displays the Principal Components ($PC$) for winter 2008/09, i.e, the coefficients obtained once projecting the data onto the PC directions. Winter 2008/09 was chosen as an example for its strong SSW event with wind reversal observed on January $24^{th}$ 2009. There is a joint sudden increase of both $PC_1$ and $PC_2$ followed by a quick decrease of $PC_2$ and a slow decrease with an inversion of anomalies in $PC_1$, i.e., the previously warm upper stratosphere is cooling much more quickly than the lower part of the stratosphere; an observation consistent with the longer radiative relaxation timescales of the lower

stratosphere (Hitchcock et al., 2013b). Similarly to the work of Hitchcock et al. (2013a), our aim is to design an index to capture such temporal evolution of the vertical temperature structure: a slow recovery, also known as Polar Night Jet Oscillation, in the lower part of the stratosphere suggests that this type of perturbation is likely to have the strongest and longest-lasting impact on the troposphere (Black and Mcdaniel, 2004; Maycock and Hitchcock, 2015; Karpechko et al., 2017b; White et al., 2020; Rao


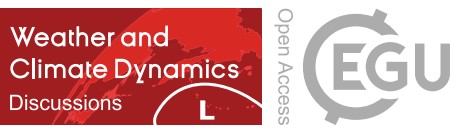

et al., 2020). Thus, better predicting SSWs with a slow recovery is expected to improve weather predictability at the Earth's
surface.

We now aim at characterizing not only the vertical structure of the temperature perturbations at individual time steps but also
their variation over time. Applying PCA to a time-lagged embedding (Takens, 1981; Sauer et al., 1991), i.e., a concatenation
of successive time steps, a combination also known as *multi-channel singular spectrum analysis* (M-SSA; Allen and Robert-
son, 1996), enables us to uncover patterns of maximum variability over fixed temporal windows. The underlying principle is
straightforward: we apply PCA not on single time steps but on larger vectors that result from the concatenation of multiple
time steps, also called time-lagged embeddings or delay-coordinate maps. More precisely, we suppose that we have a vector
$X(t_n) \in \mathbf{R}^D$ that is available for all time steps $n = 1, \ldots, N$. For M-SSA, instead of applying PCA directly on $X$, we construct
a new data set $Y$ by concatenating $T$ consecutive time steps, i.e.,

$$X(t_1), \ldots, X(t_N) \Rightarrow \begin{cases} Y(t_1) & = & [X(t_1), \ldots, X(t_{1+(T-1)})], \\ \ldots & = & \ldots, \\ Y(t_{N-(T-1)}) & = & [X(t_{N-(T-1)}), \ldots, X(t_N)]. \end{cases}$$

The length of the vector $Y(t), t = 1, \ldots, N - (T-1)$, is thus simply given by the size of $X(t)$ multiplied by the size of the
embedding window $T$. The framework developed here allows us to uncover not only vertical patterns but also their temporal
evolution.

Figure 3 shows the first four principal directions of highest variance when using a temporal embedding of $T = 60$ days. We
consider multiple values for $T$ and find two months to be most successful to retrieve meaningful and interpretable dynamics
for SSWs. As prescribed by the theoretical properties of singular spectrum analysis (Ghil et al., 2002), we observe that PC
directions come in pairs: the first two, $P_1^T$ and $P_2^T$, characterize long-lasting temperature perturbations, $P_1^T$ being in approx-
imate phase quadrature with $P_2^T$. Similarly, the third and fourth PC directions, $P_3^T$ and $P_4^T$, characterize a fast downward
progression of the anomalies. We also observe that the variations of the upper stratosphere always precede perturbations at
lower levels, which is consistent with stratospheric dynamics, where temperature anomalies are first induced by wave breaking
in the upper part of the stratosphere and then descend through wave - mean flow interaction. Figure 4 shows the corresponding
principal components obtained by projecting the data from winter 2008/09 data onto the PC directions. This event was selected
as an illustrative example both for its strength and very clear slow recovery. The 2009 SSW event is characterized by large
positive values of the third principal component around mid-February, followed by a large value of the second PC about two
weeks later: this behavior is characteristic of sudden stratospheric warmings with a slow recovery and can be used to define a
continuous scale index characterizing them (Hitchcock et al., 2013a).

Figure 4 shows for all winters the evolution of the principal components $PC_2^T$ and $PC_3^T$ obtained with the help of a PCA
on temperature anomalies with a temporal embedding of $T = 2$ months: as shown by the dashed black lines the large majority
of SSW events are characterized by a large $PC_3^T$ followed by a large value of $PC_2^T$, similar to winter 2008/09. The four
exceptions, the SSW events of the winters 1980/81, 1999/2000, 2006/07, and 2007/08, displayed by the four solid black
lines not passing through the upper right corner of Figure 4, correspond to vortex perturbations with zonal wind reversal but
very quick vortex recovery; these events cannot be considered as SSW events with slow recovery except for winter 1980/81

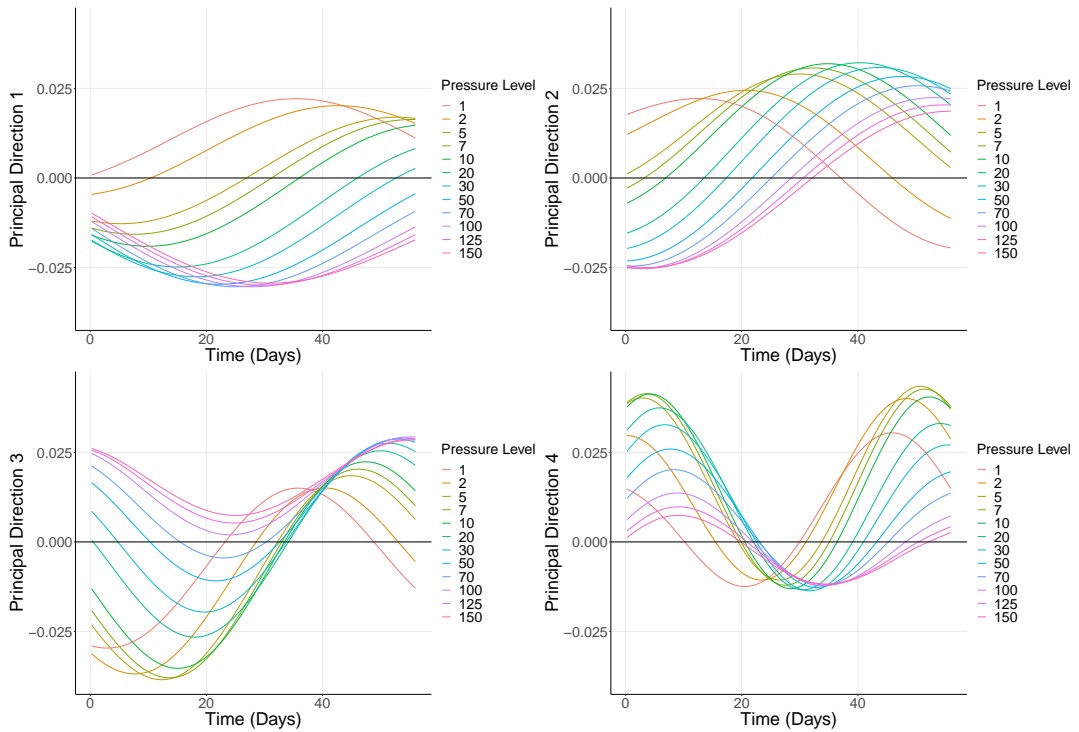

**Figure 3.** First four principal directions corresponding to the largest variance obtained from PCA on polar cap averaged temperature anomalies at levels $150, 125, 100, 70, 50, 30, 20, 10, 7, 5, 2, 1$ hPa, indicated by the different colors in the legend, with a temporal embedding of $T = 2$ months.

were the wind reversal takes place after the vortex perturbation. Figure A1 in Appendix A illustrates the temporal variation of each principal component for these four winters. Based on this analysis, we propose the following index to characterize SSWs events with slow recovery

$$I(t) = \sqrt{\left\{PC_2^T(t)\right\}^2 + \left\{PC_3^T(t)\right\}^2} \exp\left(-|\theta_{23}(t) - \pi/4|\right), \quad t = 1, \dots, N - (T-1), \tag{1}$$

where $\theta_{23}(t)$ is the direct angle between the vector $\left(PC_2^T(t), PC_3^T(t)\right)$ at time $t$ and the axis, the coefficients obtained by projecting the data set $Y$ on the PC directions $P_2^T$ and $P_3^T$ produced by a PCA with a temporal embedding of $T = 2$ months. Values of $I(t)$ are displayed by the bold black line and the color scale in Figure 4. Equation (1) is motivated by Figure 4, where SSW events with slow recovery correspond to values in the upper right corner where $I(t)$ reaches a maximum. Also, because machine learning techniques tend to focus on large values of the target variable, we design the index to be strictly positive with the largest values for SSW events with slow recovery.

We have thus managed to produce a continuous scale index using unsupervised machine learning techniques allowing us to characterize the dynamics of strong temperature perturbations of the polar vortex that are followed by a slow recovery. The index $I$ is similar to the PJO event characterization in Hitchcock et al. (2013a), but has a continuous scale, thus does not require

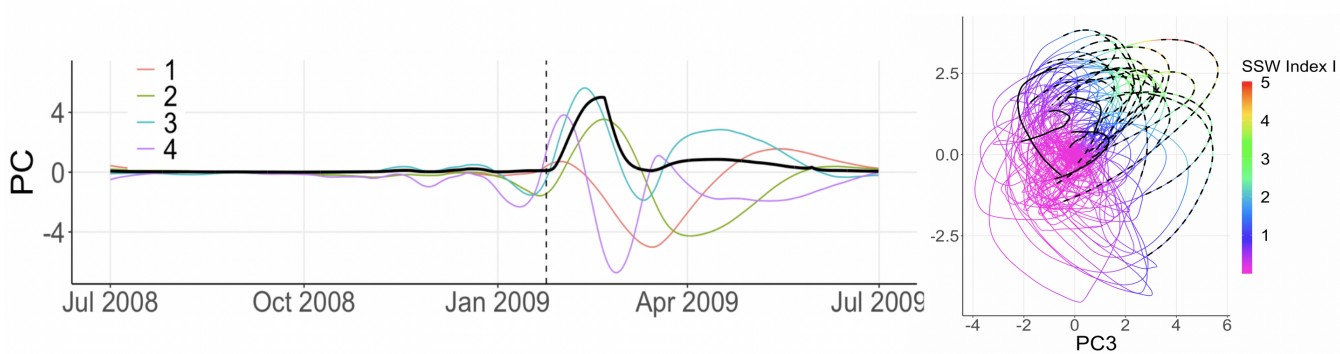

**Figure 4.** Left: First four principal components for winter 2008/09 obtained using PCA on polar cap averaged temperature anomalies at pressure levels $150, 125, 100, 70, 50, 30, 20, 10, 7, 5, 2, 1$hPa with a temporal embedding of $T = 2$ months. Vertical dashed line: central date of the 2009 SSW. Black solid line: SSW index $I$ over the same period defined as a function of the second and third principal components as in (1). Right: $PC_3^T$ against $PC_2^T$ for all winters from a PCA on polar cap average temperature anomalies with a $T = 2$ months temporal embedding. Black dashed lines: two weeks prior to and after the onset of SSW events, as defined by the zonal wind reversal criterion at 10hPa and $60°$N. Black solid lines: two weeks prior to and after the onset of SSW events, as defined by the zonal wind reversal criterion at 10hPa and $60°$N, with fast recovery. The value of the SSW index $I(t)$ is given by the color gradient.

to select any threshold and additionally explicitly characterizes the temporal evolution of the vertical temperature structure. We obtain a continuous measure of vortex perturbations that can be leveraged to efficiently apply advanced machine learning techniques.

## 4 Early signs of vortex perturbation

We now aim at uncovering relevant variables, atmospheric regions, and associated spatio-temporal patterns that are indicative of likely future SSW events with a slow recovery. We leverage supervised machine learning algorithms to predict the index $I$ proposed in Section 3. The design choice for the algorithm is determined by the following constraints: we would like first and foremost for the methodology to be interpretable, i.e., that we cannot only predict the index but also retrieve the atmospheric states yielding large index values. Second, the methodology should be able to handle large quantities of data, as we want to

search for precursors using multiple physical quantities at multiple levels and over the whole Northern hemisphere polar cap. We thus choose to apply supervised Principal Component Analysis (sPCA; Barshan et al., 2011): the algorithm computes the pattern that yields maximum correlation between the project data and a given target variable. A more classical alternative would have been canonical correlation analysis (CCA; Knapp, 1978), which is tightly linked with classical linear regression. However CCA is computationally more expensive and requires that the dimensionality of the input vector remains lower than the number

of replicates; sPCA does not suffer from such restrictions. A similar approach has been followed by Kretschmer et al. (2017) using linear regression combined with a causal discovery algorithm: their approach is efficient and provides convincing results



but cannot scale to very large problems such as ours where we jointly analyze multiple levels and, being a two-step procedure, the methodology is likely to suffer from selection bias in high-dimensional setups.

From a practical perspective, we aim at finding atmospheric patterns showing a statistically significant level of predictability, summarized here by correlation, for our index $I(t)$ at a lead time $\tau > 0$ in the future. More formally, for an input vector $X(t) \in \mathbf{R}^D$, representing any field from the reanalysis output, and a lead time $\tau > 0$, sPCA provides a solution to

$$\underset{P \in \mathbf{R}^D}{\operatorname{argmax}} \quad \operatorname{corr}\{\langle X(t_{1:(N-\tau+1)}), P_\tau \rangle, I(t_{\tau:N})\}, \tag{2}$$

where $t_i$, $i = 1, \ldots, N$ represent time steps and $\langle X(t), P_\tau \rangle$ is a dot product, i.e., the projection of the data vector $X(t)$ onto the sPCA pattern $P_\tau \in \mathbf{R}^D$ at lead time $\tau$. Equation (2) can be solved using linear algebra and inverting a, potentially large, matrix; for technical and implementation details see Barshan et al. (2011).

As a supervised algorithm, sPCA is likely to suffer from overfitting, i.e., matching noisy variations of the data instead of relevant physical dynamics. Indeed, as we explore multiple levels over a large region of the Northern Hemisphere, the dimension $D$ of the input vector $X(t)$ is meant to be large, and it is possible that the correlation between the prediction target and the data projected on a pattern $P_\tau$ might not be statistically significantly different from 0. The latter can be tested using resampling techniques: we first randomly select 10 years of data that we leave aside; this subset will not be used to train the algorithm. We then compute for each lead time $\tau$, ranging from $\tau = 1$ day to $\tau = 4.5$ months with daily increments, the patterns $P_\tau \in \mathbf{R}^D$ by application of sPCA on the 30 remaining years. Once done, we extract the patterns $P_\tau$ for each lead time $\tau$, project the data from the 10 years that were left out onto the patterns, and compute the correlation between the time series of projected coefficients and the corresponding SSW index. The train-test split procedure is commonly employed in machine learning, and we repeat it 100 times for multiple random splits to estimate the uncertainty associated with our data set. To ensure the representativeness of the testing set, the split is stratified by SSW occurrences, as defined by the zonal wind reversal criterion at 10hPa and $60°$N, i.e., we ensure that $50\%$ of the years include a sudden stratospheric warming in each set. In general, splitting is done purely at random, but in this setting, it is necessary to exclude data from entire winters to limit the impact of temporal dependence, which is likely to impact the model evaluation by artificially inflating performance metrics. We obtain a set of 100 curves describing how the estimated maximum correlation evolves as a function of the lead time $\tau$. At each individual lead time we can then assess the statistical significance at a given level of significance, e.g., $5\%$, of the correlation between the pattern projections and the SSW index by counting the number of curves with correlations close to 0; see Figure 5. More precisely, because we repeat the experiment 100 times, significance arises at lead time $\tau$ if more than 95 curves exceeds the upper bound of the estimated confidence interval under the null hypothesis $H_0 : \operatorname{corr}\{\langle X(t_{1:(N-\tau+1)}), P_\tau \rangle, I(t_{\tau:N})\} = 0$, i.e., approximately a value of $0.018$.

We start by analyzing potential vorticity: this physical quantity is available from the ERA-Interim reanalysis at 29 pressure levels from the low troposphere, i.e., 800hPa, to roughly the top of the polar vortex, i.e., 1hPa. As discussed in Section 2, directly applying sPCA to the original data would be computationally very expensive, if tractable, and susceptible to numerical instabilities, therefore we first employ the functional representation described in Section 2. Our focus being on large-scale vortex perturbations, we use only the 150 first functional coefficients for each pressure level, yielding a vector of size $D =$





$29 \times 150 = 4350$ to represent the PV field at a time $t$ over the 29 pressure levels. Before applying sPCA, we first need to normalize the PV data: a first normalization, which leads to what we simply call anomalies, consists in removing the seasonal cycle and standardizing each individual grid cell to have unit variance. The second normalization aims to assess if a pattern of planetary waves is indicative of SSWs with slow recovery and thus removes the zonal mean from the PV data at each time step.

Focusing first on anomalies, Figure 5 shows that this normalization yields the strongest correlation for short lead times, the latter being statistically significant up to 54 days in the future. We note that while the index characterizes strong SSWs with slow recovery, its value reaches a maximum about two weeks after the onset of the SSW events, as defined by the zonal wind reversal criterion at 10hPa and 60°N, so the temporal window of significance should not be interpreted as the existence of early signals about two months prior of the onset of an event, but only about six weeks before the start of the SSW event. The sPCA patterns $P_\tau$ extracted for multiple time steps can be found in Appendix B1: we observe first in Figures B1 to B4 that the predictive signal is mostly concentrated between pressure levels 1hPa to 10hPa, with no consistent pattern at tropospheric levels. During and around the peak of the SSW event, i.e., for lead times $\tau = 6$ hours from to $\tau = 14$ days illustrated by Figures B1 and B2, we observe strong circular negative anomalies centered at the pole. In Figure B3, as the lead time increases, the pattern transitions to bi-modal with negative anomalies above the Aleutian islands and positive anomalies above Europe on pressure levels 10hPa to 2hPa. Then, in Figure B4, the pattern finally returns to a circular shape with positive anomalies above the pole.

For PV waves, we analyze the atmosphere only above 500hPa as lower levels yield non-relevant patterns dominated by numerical instabilities. As shown in Figure 5, we find weaker signals with lower correlation significant only up to 48 days. Analyzing the corresponding patterns, we find that for short lead times in Figure B6, the barely significant signal is strongest in the lower part of the stratosphere with two regions of negative anomalies, one above Europe and one above the Aleutian islands, and one positive "hot spot" above Siberia. In Figures B7, B8 and B9, the predictive pattern shifts upwards in the stratosphere for lead times $\tau = 14$ days to $\tau = 60$ days, with a strong wave-1 pattern of negative anomalies above Aleutian islands and positive anomalies above Europe similarly to PV anomalies. At lower stratospheric levels, the tri-polar pattern described for shorter time horizons remains stable.

We also searched for early signals in other physical quantities of the climate system. The divergence of the Eliassen-Palm flux (EP flux; Eliassen and Palm, 1960), which is used to quantify the eddy momentum and heat transport by waves, is computed for the levels between 700hPa and 1hPa. Strong EP-flux convergence can lead to a deceleration of the westerly winds of the stratospheric polar vortex, which in case of a sufficient weakening and reversal of the winds to easterlies corresponds to a SSW (Limpasuvan et al., 2004). By application of sPCA on the divergence of the EP Flux, we found a statistically significant correlation up to a lead time of 38 days, with lower correlations than if the sPCA is applied to PV anomalies; the evolution of the correlation with time is shown in Figure 5. The pattern corresponding to lead time $\tau = 28$ days in Figure B13 reveals a strong negative heat flux at the top of the stratosphere, i.e., from 10hPa to 1hPa about which can be attributed to the breaking of planetary waves initiating the warming. Figures B11 and B12 show that the region of negative anomalies then slowly shifts downwards in the stratosphere until reaching pressure levels between 100hPa and 20hPa at a lead time of 6 hours. During its




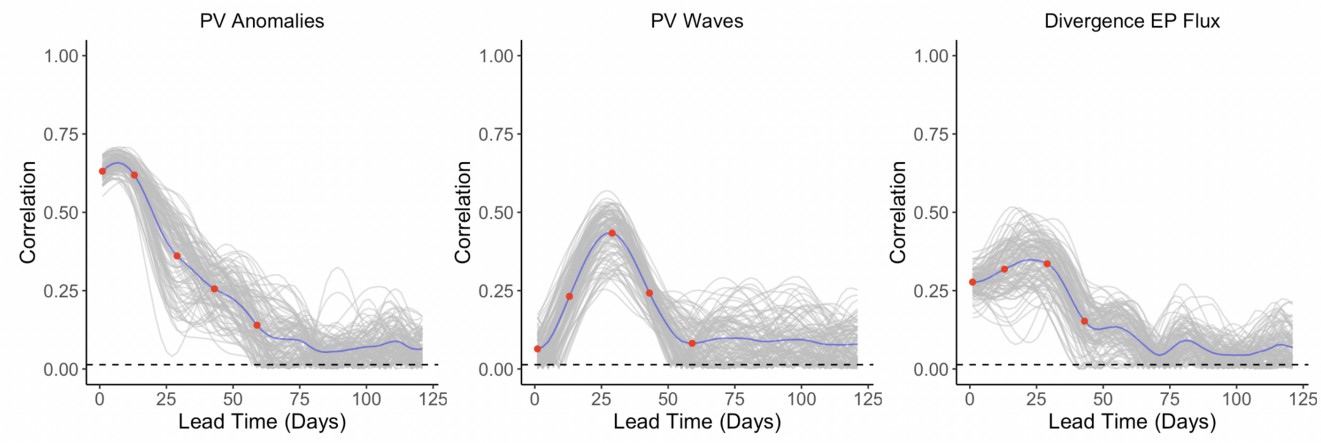

**Figure 5.** Maximum correlation as a function of the lead time for PV anomalies (left), PV waves (center) and eddy heat flux (right). Each of the gray curves is obtained by randomly selecting 30 years to which the sPCA algorithm is applied for each individual lead time $\tau = 1, \ldots, 126$ (days) to predict the index $I$. Correlation is then computed using the 10 remaining years. Random winter selection and splitting is repeated 100 times to obtain the displayed set of curves. The blue curve represents the mean of all gray curves. Correlation is significant at a 5% confidence level if less than 5 curves drop below the upper bound of the estimated confidence interval under the null hypothesis $H_0 : \mathrm{corr}\{\langle X(t_{1:(N-\tau+1)}), P_\tau \rangle, I(t_{\tau:N})\} = 0$ denoted by the black dashed line. Red points represent the time steps $-6$ hours, $-14$, $-30$, $-44$ and $-60$ days whose patterns can be found in Appendix B.

descent, the region of negative anomalies is also being replaced by strong positive anomalies, giving a strong contrast between the lower and upper regions of the stratosphere, which is characteristic of SSWs with slow recovery (Kuroda and Kodera, 2004).

Finally, we also consider tropical stratospheric winds, i.e., the zonal ($U$) wind component averaged between $-5°$ to $5°$
latitude. This variable is analyzed because the zonal winds are strongly determined by the Quasi-Biennial Oscillation, which is considered as a useful indicator for the probability of occurrence of SSWs on subseasonal timescales (Garfinkel et al., 2018). However, applying sPCA on tropical $U$ did not yield any significant correlation at any lead time.

To summarize, sPCA has allowed us to find statistically significant early signals for the occurrence of SSWs with slow recovery up to six weeks prior to the onset of the event. The signal is strongest for PV anomalies with a predictive pattern
localized in the upper part of the stratosphere, i.e., from 50hPa to 10hPa. In this case, the corresponding three-dimensional pattern retrieves the vortex pre-conditioning mechanism first mentioned in McIntyre and Palmer (1983): indeed, positive PV anomalies above the pole in the upper stratosphere indicate the presence of a strong centered vortex about 45 days before the peak of the subsequent SSW.





## 5  Performance comparison of ML and dynamical models

Now that we know which physical quantities show statistically significant correlations, as well as where in the atmosphere and how much in advance these variables are relevant, we can use this knowledge to design machine learning techniques to forecast the index $I$ defined in Section 3. We will also inquire if this kind of forecasts could be used to improve the performance of the sub-seasonal numerical prediction model ECMWF S2S.

Forecast performance being a relative notion, we need reference forecasting models against which new algorithms will be
compared. The first most natural choice is the climatological forecast, i.e., a forecast whose prediction matches the corresponding day and month of the average seasonal cycle. This forecast is usually seen as the least informative as this strategy does not take into account the current state of the system; it is thus seen as the baseline that has to be beaten for a forecast to exhibit any kind of predictability.

A more informed, and more accurate, alternative, corresponding to the current state-of-the-art, are the sub-seasonal forecasts
produced by numerical models. In particular, we analyze the performance of the S2S forecasts produced by the ECMWF presented in Section 2.

Data-driven alternatives such as machine learning algorithms by default provide point forecasts, i.e., not an ensemble but only one value for each lead time. Comparing probabilistic forecasts, such as the ECMWF ensemble members, with point forecasts is therefore challenging, but turning machine learning algorithm into probabilistic forecasts is not straightforward
in general. To ensure the fairest possible comparison between point and ensemble forecasts, we use the mean absolute error (MAE) as the metric of performance: for a point forecast, the MAE is simply the absolute value of the difference between the ground truth, in this work the ERA-Interim data, and the forecast. For the ECMWF hindcasts and for the climatology, we consider the mean of the absolute difference between all equally likely outcomes and the ground truth.

To forecast the index $I$, we consider two machine algorithms as potential competitors for our reference scenarios: the first is
a simple linear regression using as predictors the projection of ERA-Interim data on the patterns described in Section 4. This approach is similar to Kretschmer et al. (2017), where instead of using sPCA to uncover relevant regions and variables, the authors rely on a causal discovery algorithm. Secondly, we used a multilayer perceptron (Friedman et al., 2009, p. 391-395) taking raw PV anomalies over multiple levels and a fixed temporal window. More precisely, we choose a neural network with three fully connected layers with 100 neurons per layer and combine them using ReLU (Rectified Linear Unit) activation functions.
Alternative architectures have also been considered and could be further fine-tuned, but only for relatively minimal performance improvement. Finally, we also considered kernel analogs resampling techniques (McDermott and Wikle, 2016; Yiou, 2014), which generate new random trajectories by re-combining previously observed atmospheric states, thereby naturally producing probabilistic forecasts. We however did not manage to successfully apply analogs techniques so that they provide any kind of performance improvement over numerical forecasts. These algorithms suffer from the curse of dimensionality, so we
attribute the poor performance to the lack of a low dimensional and sufficiently relevant representation of the vortex dynamics to measure the "distance" between trajectories. We still believe that these techniques have very attractive properties and can successfully be used to improve prediction performance in this context; we however leave such developments for future work.





To enable a comparison with the ECMWF hindcasts, we first repeat the analysis described in Section 3, and compute the index $I$ over the full period, using daily mean temperature averages for pressure levels 100hPa, 50hPa and 10hPa only. The

SSW index produced with this more limited quantity of data behaves similarly to the original one; it is thus used as ground truth to quantify forecast performance. Next, to assess and compare the performance of each machine learning algorithm employed, the reanalysis data is divided into three independent subsets: first, the validation set is used to perform parameter tuning and model selection and includes four randomly selected winters between June 1979 and June 2015; all remaining winters over this period form the training set. Finally, we use the last four winters, October 2015 to April 2019, as a test set, i.e., the data from

these winters is used to compute the MAE and to compare models performance. Note that all sets, i.e., the train, validation and test, are designed to include an equal proportion of winters with and without SSWs with slow recovery. Figure 6 gives the evolution of the MAE for each model as a function of the lead time. We first observe that all models can be considered as skillful as they provide a lower MAE than the climatological forecast. For short lead times, i.e., from 1 day to about 25 days, the ECMWF model performs best as the influence of the initial conditions is still strong. Beyond this time horizon, the MLP

starts to outperform the ECMWF prediction system, followed by the sPCA linear regression about 10 days later. The results show that ML algorithms are capable of outperforming numerical models for extended-range forecasts, and could be leveraged to improve the performance of sub-seasonal forecasts.

Therefore, we propose to post-process S2S forecasts based on ML forecasts to improve the overall model performance: for a given initialization date, we start by producing an ML forecast for $I$ at all lead times using the machine learning algorithm

with average best performance, i.e., the multilayer perceptron. We then compute the distance between each of the 11 ensemble members and the vector of predictions. Out of these 11 trajectories, we select the one with the lowest distance over all lead times, i.e., the most likely ensemble member with respect to the ML prediction; the latter is then labeled as the post-processed S2S. The presented procedure relies only on the output of the ML model and thus does not use any information that would not be known a priori at the time of initialization of the numerical model. We repeat this process for all initialization dates

of the ECMWF hindcasts to retain only one ensemble member per date. Computing distances between ensemble members and the vector of predictions using only lead times between 37 and 47 days, where the ML algorithm performs best, provides better overall performance. Figure 6 shows that the post-processed ECMWF hindcasts have an MAE up to 20% lower than the original ensemble with significant performance improvement after day 25. While the post-processed ECMWF model has a larger MAE than the original MLP algorithm for lead times of 28 to 47 days, it has the advantage of providing predictions not

only for the index $I$, but also all other atmospheric variables as we only select one of the 11 ensemble members of the ECMWF model.

We showed here that ML methods can be used to improve long-range forecast performance in the stratosphere. Further fine-tuning of different ML models, by trying more combinations of variables or hyperparameters, is likely to further improve the performance of the ML models and the post-processed S2S model, but is left for future work.

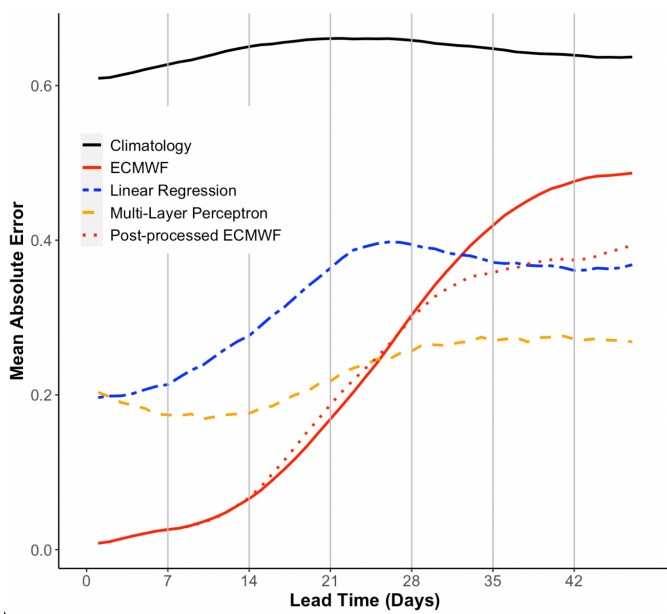

**Figure 6.** Mean Absolute Error (MAE) as a function of the lead time for the following forecasting models: Climatology (solid black), ECMWF model (solid red), linear regression with sPCA output (double-dashed blue), Multilayer Perceptron (dashed orange) and post-processed ECMWF model (dotted red).

## 6 Conclusions

We presented in this work a framework to analyze and predict atmospheric dynamics using machine learning. The methodology presented is a three step procedure: we first use unsupervised machine learning techniques to produce a univariate index quantifying the occurrence and the strength of sudden stratospheric warmings with slow recovery. The index is then used as input for supervised algorithms in order to assess "where" and "when" in the system we can find relevant predictors. Finally, the answer to these questions this knowledge is used to produce ML forecasts up to 47 in the future for the proposed SSW index. The performance of these forecasts are compared against the state-of-the-art ECMWF numerical prediction model: the latter performs best for short- to medium-range lead times of up to 25 days, while the data-driven model outperforms the dynamical prediction model for longer lead times. Machine learning forecasts are then used to post-process S2S predictions, yielding a 20% decrease in terms of mean absolute error. A part of the increased predictability at longer lead times comes from the fact that SSW events often occur as part of PJO events. Their onset in the upper stratosphere tends to occur on average slightly before SSW events, hence the onset of an SSW event that occurs during a PJO event will likely be more predictable as the PJO event has already started at the time of the SSW onset.

The methodology presented in this paper has been developed to ensure both tractability and interpretability of the results. As the drivers behind SSWs with slow recovery are large-scale circulation patterns, we successfully reduced the data dimensionality using a functional representation of the data that allows us to apply machine learning algorithms using reasonable




computational times and resources, making possible to test statistical significance using re-sampling techniques. We focused here mostly on algorithms such as linear regression whose interpretability is straightforward and which not only improve the forecast performance but also potentially allow us to deepen our understanding of SSW dynamics. However, nonlinear and more complex data-driven methods such as Laplacian eigenmaps (Belkin and Niyogi, 2003) or diffusion maps (Coifman et al.,

2005; Coifman and Lafon, 2006) that have already been applied to study climate dynamics (Bushuk et al., 2014; Székely et al., 2016) could also be used as replacement for PCA in Section 3. Similarly, in Section 4, using the kernelized generalization of sPCA could also help uncover non-linear relationships between potential predictors and SSW events. The linearity of sPCA might indeed be one of the reasons why our method was not able to detect any significant signal from tropospheric planetary waves. However, not all SSWs have anomalous tropospheric precursors beyond a sufficiently large tropospheric wave forcing,

and anomalies in the wave flux (Birner and Albers, 2017) or wave amplitude (Domeisen et al., 2018) often only emerge within the stratosphere, which might be a further reason why the model does not detect anomalous tropospheric precursors. Section 5 is an exception to our constraint of interpretability as we considered a multilayer perceptron as potential candidate for forecasting model: neural networks are difficult to interpret in general, but their interpretability is a very active field of research (e.g. Lundberg and Lee, 2017).

The major drawback of the post-processing presented in Section 5 is that the probabilistic interpretation of ensemble dynamical forecasts is lost by selecting only one relevant trajectory. Potential improvements would thus refine the processing by computing mixtures weights, i.e., the relative likelihood of each ensemble member with respect to the current forecast. To achieve this goal, possible directions could be using e.g. kernel weights proportional to the distance between the numerical and the ML forecasts or more generally producing probabilistic ML forecasts. In the latter, the likelihood of each ensemble

member could then be deduced directly from the distributional forecast. As mentioned in Section 5, usage of kernel analogs was attempted but without success. A deeper understanding of the vortex dynamics and its representation in low-dimensional spaces will be essential to produce analogs that outperform numerical models on all timescales.

*Code and data availability.* ERA-Interim reanalysis was obtained from the ECMWF server (https://apps.ecmwf.int/datasets/data/interim-full-daily). The S2S data are publicly accessible at https://apps.ecmwf.int/datasets/data/s2s.

*Author contributions.* Raphaël de Fondeville designed, implemented and conducted all the data analysis with input from the co-authors. He also made the figures and wrote the manuscript draft. All authors contributed to interpreting the results and editing the manuscript.

*Competing interests.* The authors declare no competing interests for this project.



*Acknowledgements.* This work was funded through the project *EXPECT* (C18-08) of the Swiss Data Science Center. Partial support from the Swiss National Science Foundation through project PP00P2_170523 to Z.W. and projects PP00P2_170523 and PP00P2_198896 to D.D. is gratefully acknowledged.






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





## Appendix A: Dynamics of SSW events without slow recovery

**Figure A1.** First four principal components for winters 1980/99, 1999/2000, 2006/07 and 2007/08 obtained using PCA on polar cap averaged temperature anomalies at pressure levels 150, 125, 100, 70, 50, 30, 20, 10, 7, 5, 2, 1hPa with a temporal embedding of $T = 2$ months. Vertical dashed line: central date of the 2009 SSW. Black solid line: SSW index $I$ over the same period defined as a function of the second and third principal components as in (1).





## Appendix B: Three dimensional patterns of SSW early signals.

### B1 PV Anomalies

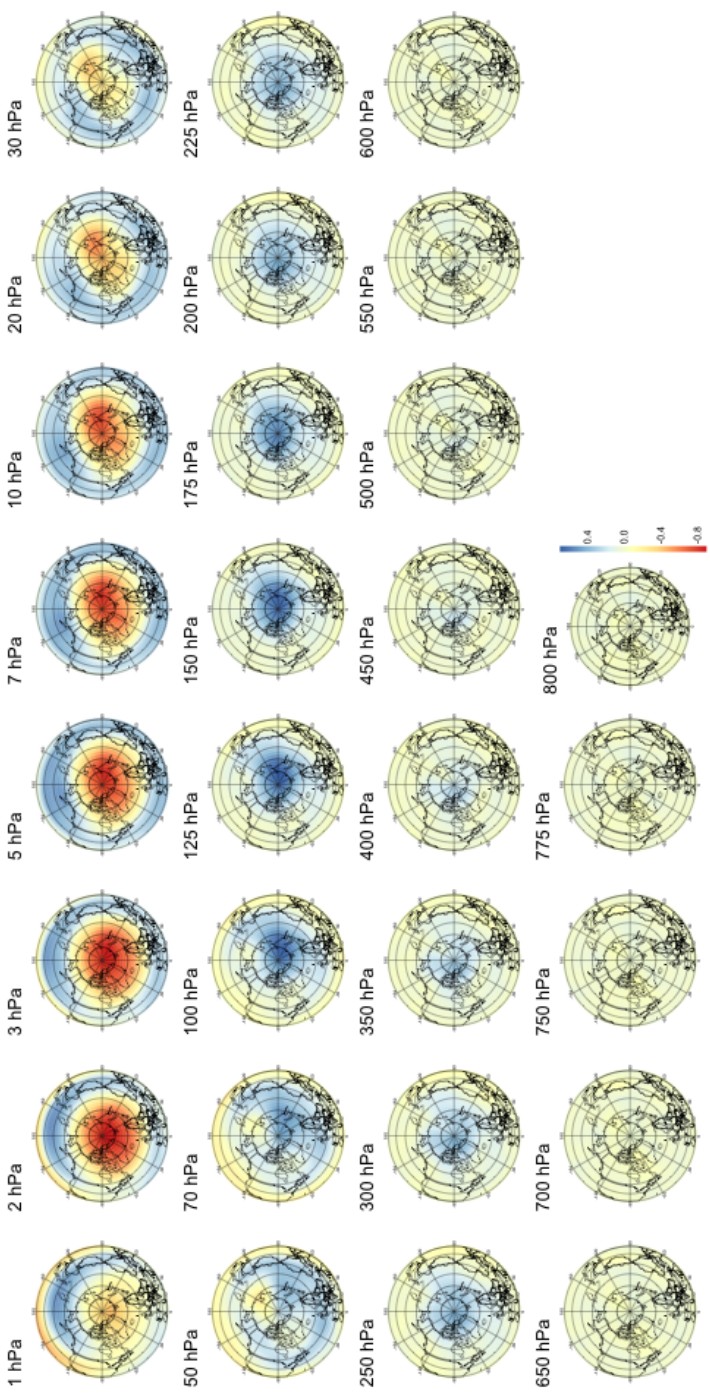

**Figure B1.** Pattern of PV anomalies maximizing correlation with the SSW index at a lead time of 6 hours.

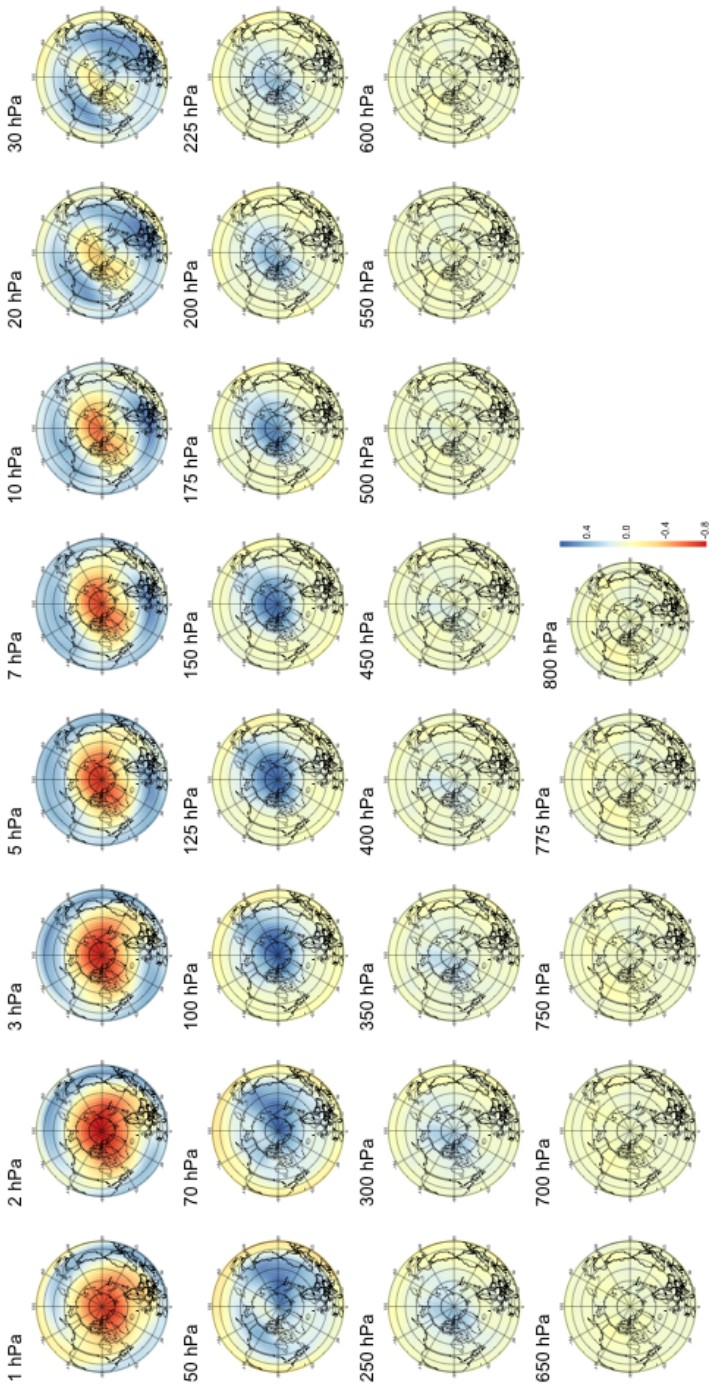

**Figure B2.** Pattern of PV anomalies maximizing correlation with the SSW index at a lead time of 14 days.

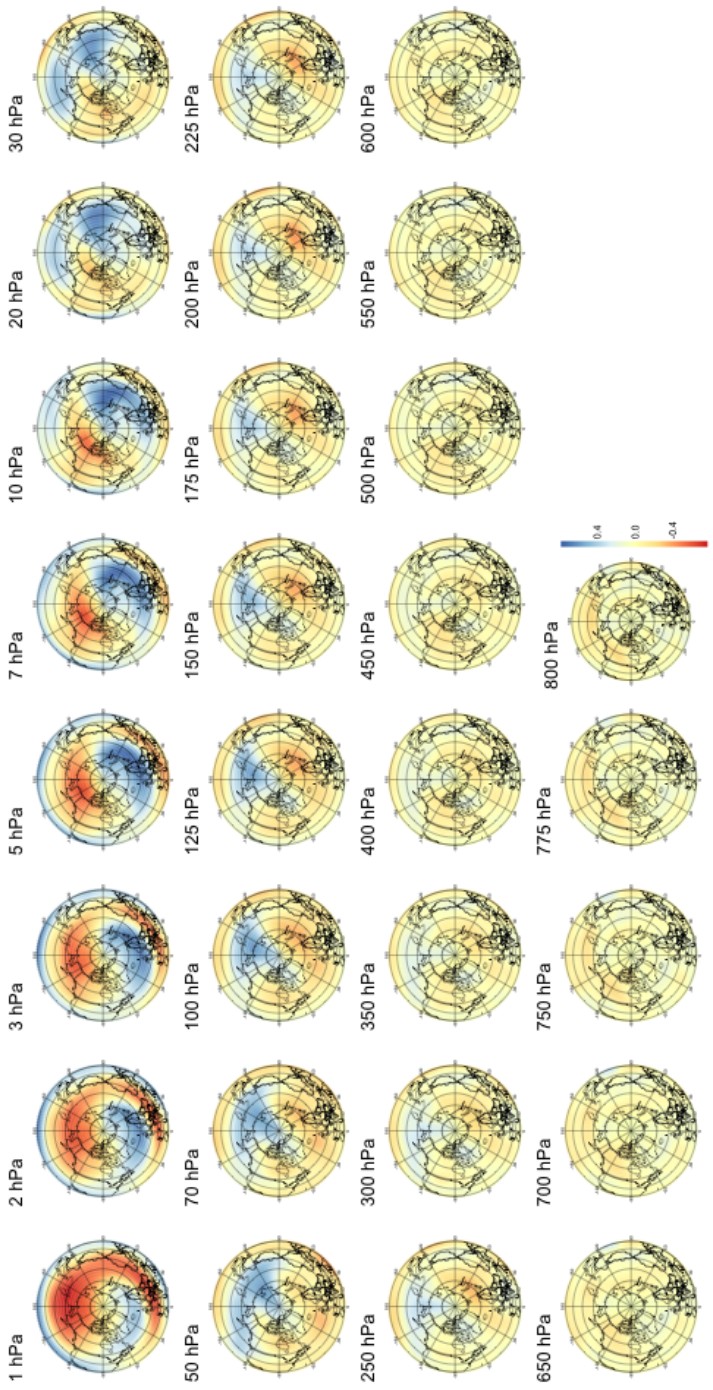

**Figure B3.** Pattern of PV anomalies maximizing correlation with the SSW index at a lead time of 30 days.



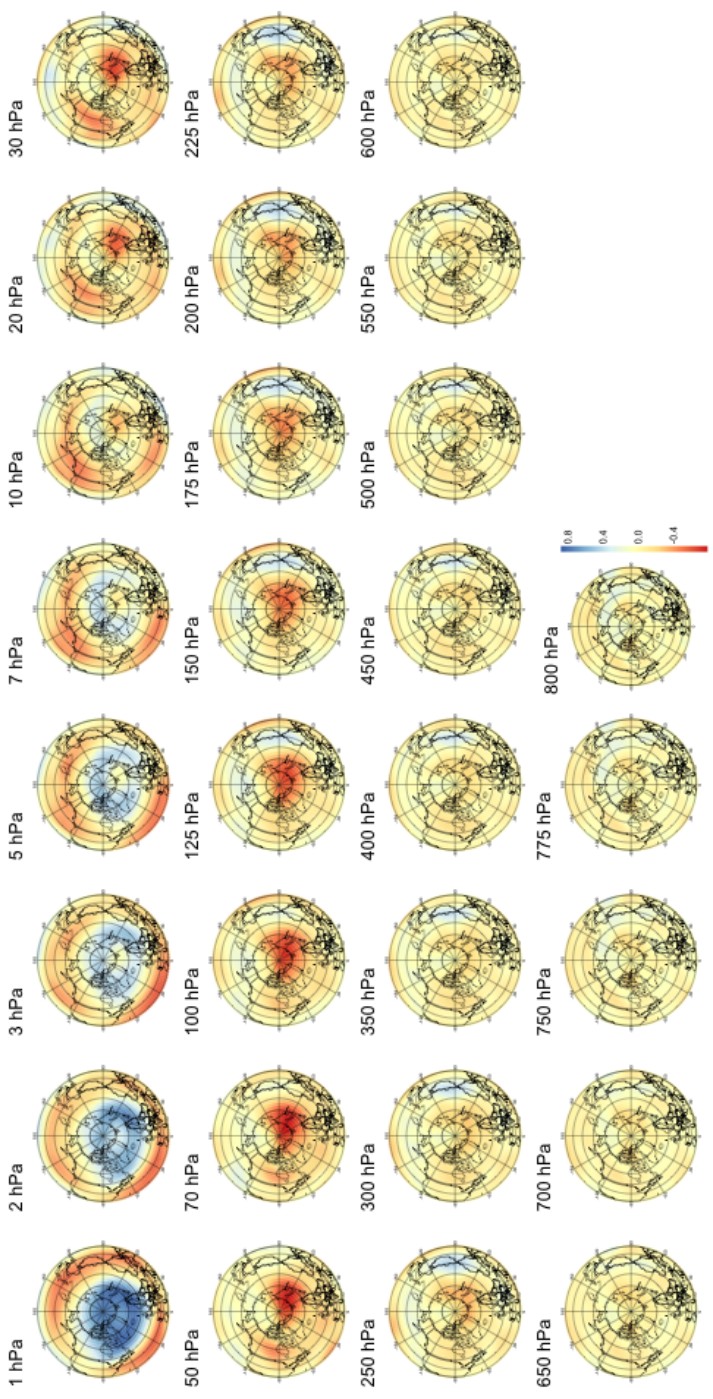

**Figure B4.** Pattern of PV anomalies maximizing correlation with the SSW index at a lead time of 44 days.

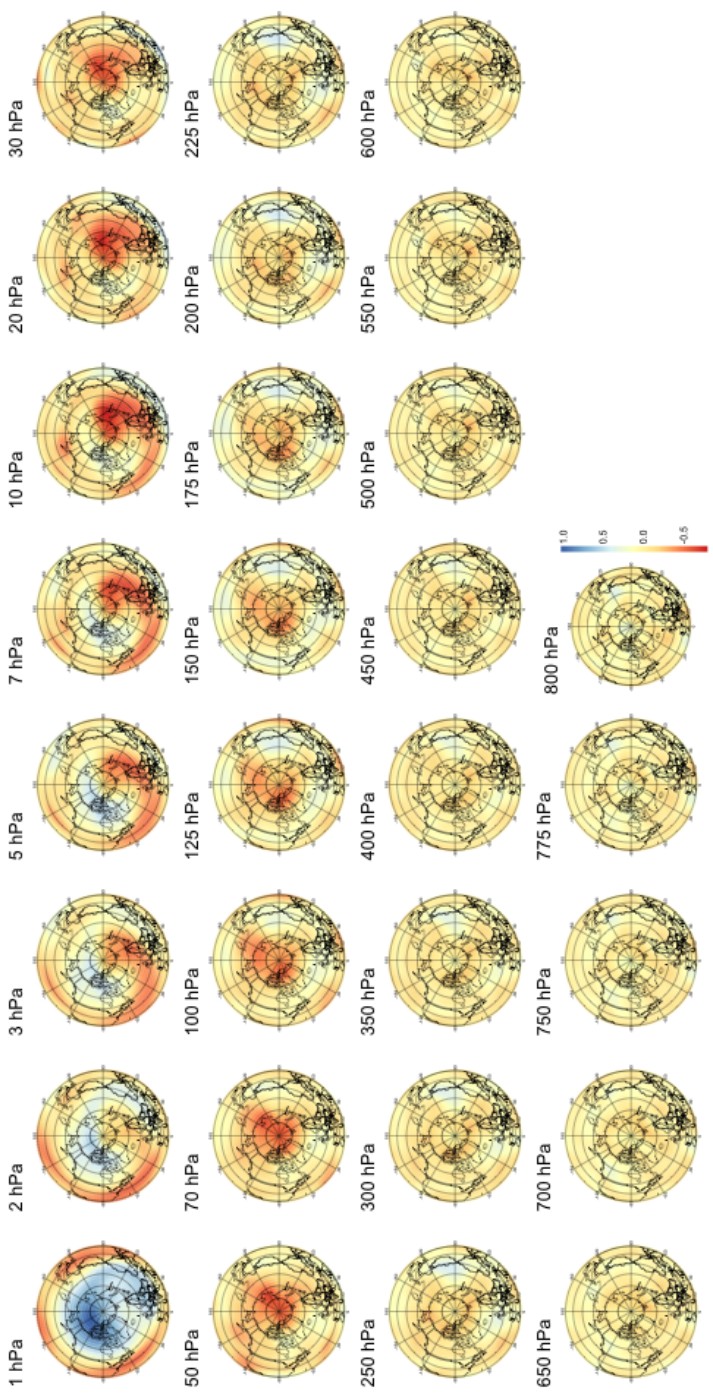

**Figure B5.** Pattern of PV anomalies maximizing correlation with the SSW index at a lead time of 60 days.





## B2  PV Waves

560

**Figure B6.** Pattern of PV waves maximizing correlation with the SSW index at a lead time of 6 hours.



**Figure B7.** Pattern of PV waves maximizing correlation with the SSW index at a lead time of 14 days.

**Figure B8.** Pattern of PV waves maximizing correlation with the SSW index at a lead time of 30 days.





**Figure B9.** Pattern of PV waves maximizing correlation with the SSW index at a lead time of 44 days.



**Figure B10.** Pattern of PV waves maximizing correlation with the SSW index at a lead time of 60 days.





## B3   Heat Flux

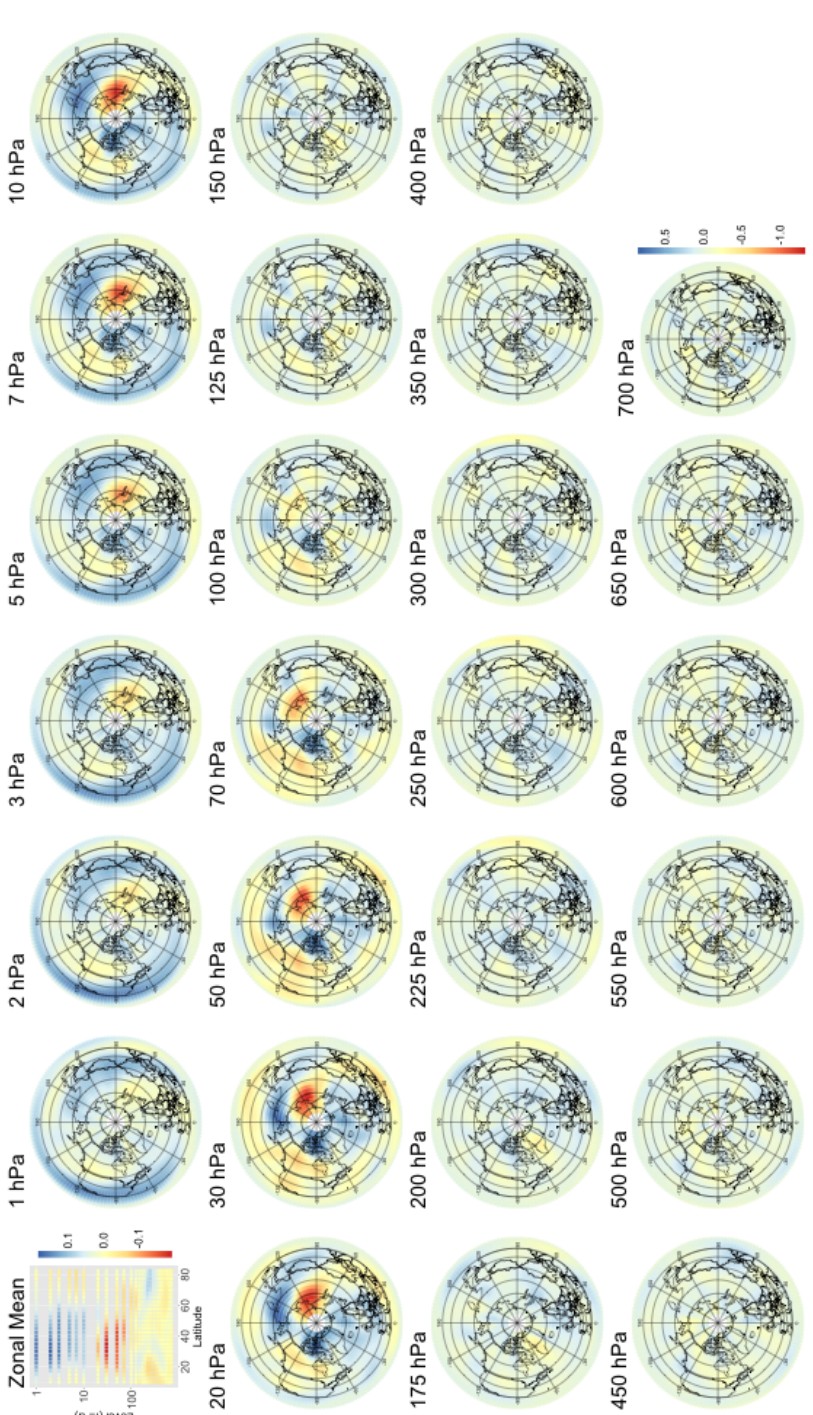

**Figure B11.** Patterns of heat flux maximizing correlation with the SSW index at a lead time of 6 hours. The top row, first column plot displays the patterns zonal averages.



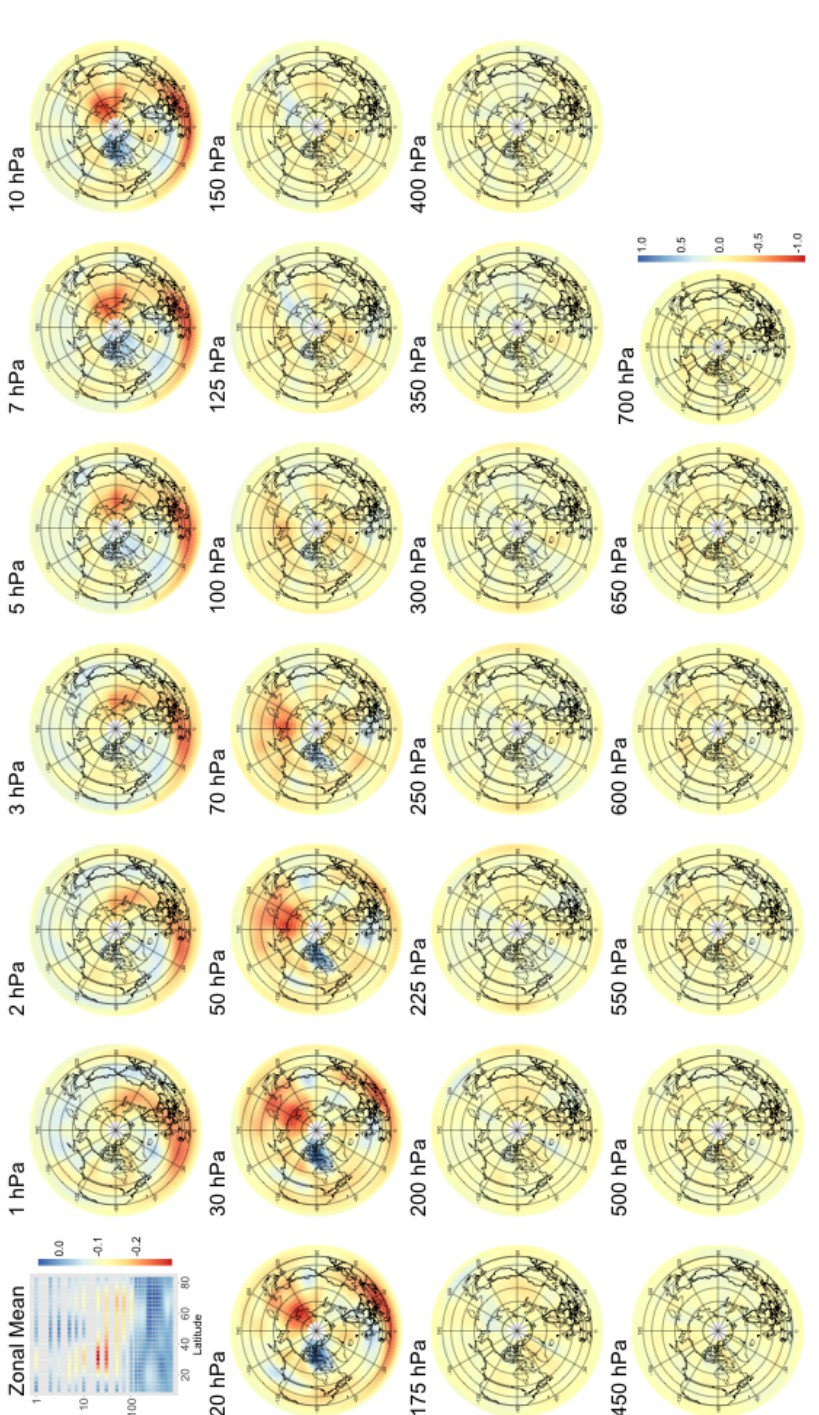

**Figure B12.** Patterns of heat flux maximizing correlation with the SSW index at a lead time of 14 days. The top row, first column plot displays the patterns zonal averages.

**Figure B13.** Patterns of heat flux maximizing correlation with the SSW index at a lead time of 30 days. The top row, first column plot displays the patterns zonal averages.

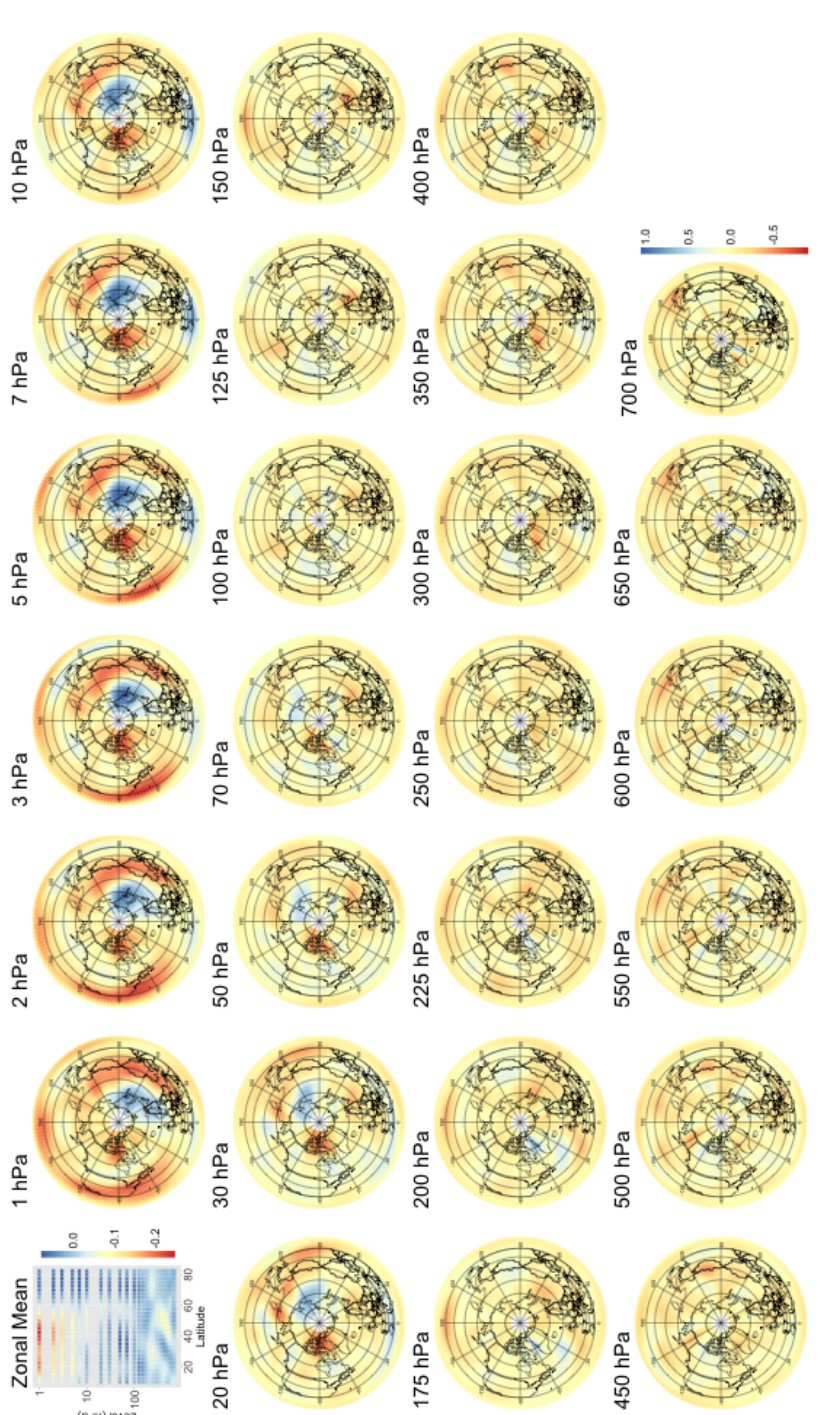

**Figure B14.** Patterns of heat flux maximizing correlation with the SSW index at a lead time of 44 days. The top row, first column plot displays the patterns zonal averages.