# Peer review of "Improved Extended-Range Prediction of Persistent Stratospheric Perturbations using Machine Learning"

_Weather and Climate Dynamics, 2022_

## Author Response (AR1)

**wcd-2022-55 - Response to Reviewers**

First of all, the authors would like to sincerely thank the reviewers for having carefully read their manuscript and made such valuable comments. Thanks to their help the quality of the publication has without doubts been improved. All modifications are highlighted in red in the revised version of the paper.

**1 Reviewer 1**

• My first and primary concern is with the data-driven index for slowrecovering SSWs. My understanding is that you opted not to use a standard definition, e.g., the reversal of zonal-mean zonal wind at 60N and 10hPa as per the WMO, for two reasons: (1) to have a continuous sliding scale, and (2) to avoid arbitrariness. However, you could easily modify the WMO definition to make it continuous, e.g., by changing the zonal-wind threshold or the duration. Also, standard definitions enjoy the significant advantage of being simple and objective, whereas data-driven methods such as the one you describe (PCA on temperature profiles) seems to require far more arbitrary choices, such as the size of the polar cap region, the pressure levels, and the spatiotemporal resolution. You do back up your choice by citing multiple preceding studies linking the temperature profile with surface impacts, so I'm not asking you to change the procedure, but the standard definition could be dealt with more thoroughly. I would hope that many key conclusions of the study, such as statistically significant precursors, would not change much if you were to swap in the standard definition. However, you do refer to the standard criterion at various later points in the paper; overall, I was unsure to what extent the WMO definition was being used as a parameter or just a point of comparison. Is a large value of your perturbation index, I, highly correlated with the occurrence of an SSW defined by the WMO? Some comments about this would be welcome.

We thank you very much for such a detailed comment. First, to clarify, the WMO definition is used as a point of comparison and not as a parameter in this work: we are aware that our index differs from the classical and highly interpretable WMO definition, that is why we wanted to provide some elements of comparison in Section 3.

The search for an alternative characterization was motivated by the work of Coughlin and Gray (2009), who find that there is a continuum of SSWs instead of one unique category. For this reason, we wanted to use a datadriven methodology on the temporal and vertical dynamics of the temperature to characterize such a continuum. This led us to further distinguish SSWs into SSWs with and without slow recovery; the latter type being often closer to vortex deceleration events. We thus developed the index Iwhose large values correspond to SSWs with slow recovery, which are related to polar jet oscillation events. This category of SSW is of particular importance as these events are likely to have a long-lasting influence on the weather at the Earth's surface and make them the best candidates to improve forecast performance.

Concerning the correlation, from the perspective of the physical process, the two indexes are meant to be highly correlated as our index *I* characterizes the large majority of SSWs, i.e., the ones with slow recovery. Indeed, only a small subset of SSWs are followed by quick recovery of the vortex. However, from a mathematical point of view, estimating this correlation is delicate to compute as both definitions are not aligned temporally: while the WMO definition can be encoded thanks to a binary variable taking the value 1 at the date of the first wind reversal and 0 elsewhere, our index takes continuous positive values and reaches a maximum about two weeks after the central date of the SSW, i.e., about half-way through the recovery stage. For this reason, a simple correlation between the two indexes might be low even though they both characterize similar events.

• Figure 3 can use some clarification. First of all, I didn't see the amount of variance in the first four principal directions reported anywhere—did I miss it? This seems like relevant information. Second, I found the colors and overlapping curves to be hard to parse. The high- and low-pressure levels are colored quite similarly, and the "fast downward progress of the anomalies" (c. line 176) is not so obvious to me. A heat map might be more clear, with each row corresponding to a pressure level and each column a time sample. That is how I'm used to visualizing downward propagating anomalies, e.g., as in Baldwin & Dunkerton (2001) Fig. 2. Third, it is hard to connect mixtures of these principal directions with a pattern of vertical temperature change.

These are very good points thank you very much. We have added the corresponding explained variances in the caption of each figure. We also changed Figure 3 to turn the graphical representations into heatmaps.

• What is the reason to choose sPCA instead of linear regression (perhaps with regularization to promote sparsity, e.g., LASSO)?

The motivation behind the usage of sPCA over linear regression was that we needed a methodology that would be applicable in a potentially highdimensional setting (input vector larger than the sample size). With this motivation alone, we could also indeed have considered using a LASSO or a ridge penalty. However, we preferred sPCA as it also offers a kernelized version that could be used to go beyond linear relationships. This was mentioned in the conclusion, but for better clarity we added a sentence in Section 3 along with the suggestion to use the LASSO/Ridge regression as an alternative for linear dependencies.

• Fig. 4: Some material is cut off at the interface between the left and right panels. For example, the vertical axis should have a "PC2" label, whereas the "Jul 2009" axis tick label is cut off. Further, because the PCs are not standard, it was not obvious (at least not to me) which direction to follow the black curves in the right-hand panel. I trust they all move in the same direction (counterclockwise, as suggested by the text)?

We fixed all these issues and added missing details as you suggested. Many thanks for spotting them.

• line 175 (time-delay embedding notation): there is some mixing between the subscripts on time  $(t_1, ..., t_N)$  and the time itself (t = 1, ..., N - (T-1)). It would be better to stick to one notation for simplicity.

Thanks, there was indeed a possibility for confusion. The new version should be clearer.

**2 Reviewer 2**

• ML forecast using a deep neural network (DNN): In general your work impressively outlines, a reliable application of machine learning techniques in research, as you leverage physical knowledge to present a concise learning task to the network. However, for me some questions and concerns remain. First, the MLP forecasting is not reproducible, due to the very limited description of the training parameters, input and output dimensionalities and other specifics. Even though to me this is not necessarily main body material, including the information in the appendix or supplementary material (maybe even code or a trained model) is necessary for reproducibility. My second concern is the lack of statistics for the results of the MLP. DNN results sacrifice reliability due to cherry picking, i.e. training only one network. I suggest, for example retraining the model several times given different initial parameters, i.e. deep ensemble approach, yielding more substantial results. In addition, this procedure provides you with the lacking ensemble information, discussed in line 318-320. Lastly, such ensemble approaches as well as the field of Bayesian Learning, I would reframe the statement circa line 317, where you address the default of point forecast for machine learning algorithms, as it is only partly true. Also, I would suggest adding the according citation that supports your argument in line 318-320.

Thank you very much for raising these points that indeed would benefit from further details. We first would like to stress that we are not using a DNN but a simple and "small" NN with 3 fully connected layers of size 100 connected with ReLu activation functions. It is thus a rather small and simple network. The model was implemented using the 'torch' package from the CRAN repository and optimized thanks to an Adam optimizer with a learning rate of 0.001. We have added details on the optimizer and on the dimension of input vector in Section 5. The output is simply the univariate index value at the desired lead time  $\tau$ .

The details above might clarify your second point: as you can see our MLP is neither deep nor large. For this reason, we observed that, for a given architecture, models with different initialization points were consistent with each other. Also, as we got convincing results with such a simple architecture, we did not do extensive optimization of the architecture, i.e., there was no form of cherry-picking as we tested only a limited number of architectures, up to 4 layers, with sizes ranging from 10 to 200: all models yielded either no significant difference in performance or a drastic performance loss for too "small" networks.

Using an ensemble of MLP models could indeed provide some form of variability, however the interpretation of such variability is quite unclear and not comparable to the uncertainty of numerical models. That is why we preferred, at least for now, to not consider such a direction. However, you are indeed right about the possibility to leverage Bayesian Neural Networks, so we now refer to this possibility, which we leave for future work as already mentioned in the Conclusion. We also add a word on ensembling as an attractive research avenue in this paragraph. The sentence reads as follow "Modifying machine learning algorithms to output probabilistic forecasts is possible but requires either advanced techniques such as Bayesian computation or models ensembling, whose link with numerical ensembles is still not well understood. Thus, in this exploratory study, we focus on classical ML algorithms, leaving probabilistic modeling for future work.".

• Post-processing Equation: Overall, the description of calculation procedure as well as detailed equations help to understand and reproduce the results. The improvement I want to suggest, concerns the post-processing used to enhance the numerical ensemble forecast. I think combing existing descriptions with an equation would complete the paragraph.

Thanks for the suggestion we have added such an equation.

• Visualization: While the figures in your work provide strong and straight forward visualisation, especially Fig. 3 can profit from improvements, as well as the discussion of Fig. 5. In terms of Fig. 3, I agree with the Comment (RC1) of Reviewer 1 and have nothing to add. Regarding, Fig. 5, during the discussion of the results you do not specifically mention which of the three plots you address, which sometimes makes it hard to follow the conclusions. Thus, some of the visual arguments do not become evident

right away. My recommendation is either a more descriptive wording or a more distinct visualisation.

Thank you for the suggestions. Figure 3 was changed to heatmaps as suggested by RC1.

For Figure 5, we have improved the wording in the text. We hope that it is now clearer.

• Citation of Kretschmer et.al 2017: In line 220-223 you refer to a publication by Kretschmer et. al. and discuss the inability of the algorithm, put forward in this paper, to scale to large problems. What do you mean by the selection bias and can you clarify why the algorithm does not scale to large problems?

Selection bias: the algorithm in Kretschmer et.al 2017 requires to perform a statistical test for each coefficient of the linear regression. We are thus in a multiple testing setting and the algorithm should be modified accordingly (not done in their work). The difficulty is that with a potentially large dimensionality of the input vector, a Bonferroni correction is likely to be inconclusive in general. Without proper treatment of multiple testing, which is not trivial, the number of potential "false positives" is likely to be large causing a selection bias.

Scaling: apart from the above issue, the causality algorithm relies on a linear regression, thus if the input vector size is larger than the number of observations, it cannot be applied; see the response to comment 3 of RC1. In our case, we have a gridded product over multiple levels and multiple time steps (and we could also consider multiple fields at the same time). So, it is likely that the dimensionality of the input vector increases to a size where their algorithm is not applicable (the matrix is not invertible).

• Technical Corrections:

Thank you for spotting all these technical issues. We have implemented them all as best as we could: the plots were adjusted following your suggestions, and we have added the word "learning" were it was missing.

---

## Referee Report (RR1)

**Review: WCD 2022-55**

**Title:** Improved Extended-Range Prediction of Persistent Stratospheric Perturbations using Machine Learning

**Authors:** Raphaël de Fondeville[1], Zheng Wu[2], Eniko˝ Székely[1], Guillaume Obozinski[1], and Daniela I.V. Domeisen

**General Quality:**

In general the revision improved the quality of the paper, which I suggest be accepted subject to technical corrections. The authors resolved all specific comments in my review and improved the visualisation quality.

With the final edits, this work is an important contribution to the scientific progress of S2S forecasting and demonstrates reliable integration of physics-based machine learning into climate research. The presented three step procedure provides novelty in that it combines existing data-driven machine learning techniques with current physical research findings to enable a more extensive assessment of SSW dynamics. Moreover, the procedure facilitates increasing the performance of numerical ensemble forecast for lead times above 25 days.

Overall, each research step in this work is thoroughly motivated and discussed, enabling reproducibility. The authors present strong results with according statistics (Cross-Validation) to support their conclusions.

**Technical Corrections:**

1. *ML forecast using a deep neural network (DNN):* Although, the added sentences (l.322-325) regarding Bayesian NNs and deep ensembles provide the appropriate background, I still suggest refraining from strong statements, i.e. 'whose link with numerical ensembles is still not well understood. ' without providing respective citations. I suggest adding context literature and maybe softening the statement as it is a topic of ongoing research. See for example [Abdar et. al. 2021] (https://www.sciencedirect.com/science/article/pii/S1566253521001081), who discuss the topic in detail also theoretically referring to an MLP with one hidden-layer such as yours.

2. *Visualisation:* While Figure 3 has undergone major improvements, unfortunately you have now dropped the PC labels for the individual plots. I suggest adding PC labels or adding A,B,C,D in the Figure and according assignment (A is PC1, etc.) in the caption. Otherwise, all visualisation adjustments are satisfactory

3. *Citation of Kretschmer et.al 2017:* Thank you for the clarification. I think that the statement of the authors is not completely true. The approach described in Kretschmer et al. discusses multiple testing and accounts for it as described in the SI (p2, Robustness of the causal precursor detection scheme): *"Note that the causal discovery algorithm involves multiple hypotheses testing such that the significance parameter α should be considered as a hyper-parameter of the algorithm. As a conservative confidence estimate for a predictor (which is not really relevant for prediction), one can use the maximum p-value among all tested conditions in the PC algorithm. We also tested whether the obtained predictors are significant if we additionally control the false discovery rate (FDR) and found the lag-1 SPV index and v\*T\*100 over Eurasia (Fig. 3a) to be still significant (adjusted $p <0.01$) but the other two region (Fig. 3a) slightly dropping in significance (adjusted $p \approx 0.15$)."* Thus, I suggest softening the statement but do not see this as a requirement for the publication of this manuscript.

---

## Author Response (AR2)

**wcd-2022-55 - Response to Reviewers**

The authors are extremely grateful to both reviewers for having carefully read a second time their manuscript. Thanks to their help some typos and imprecision were corrected increasing the quality of the publication. All modifications are highlighted in red in the revised version of the paper and Figure 3 has been updated following the reviewers comments.

**1 Reviewer 1**

- The authors have responded seriously and thoroughly to all of my comments. Figure 3 in particular is much more illuminating. However, this has created one new confusing point: the time-delay embedding is said to contain N = 60 lags, but the horizontal axes in Figure 3 stretch for over 200 days. This is in contrast with the first version of Figure 3, which spanned only 60 days. Has the lag time increased in the new draft, say, to 240 days?

  Thank you so much for spotting the mismatch! It is still 60 days but Figure 3 axis was showing 'time steps' (every 6 hours) instead of actual number of days. We have fixed this issue in the new version.

**2 Reviewer 2**

- 1. ML forecast using a deep neural network (DNN): Although, the added sentences (l.322-325) regarding Bayesian NNs and deep ensembles provide the appropriate background, I still suggest refraining from strong statements, i.e. 'whose link with numerical ensembles is still not well understood. ' without providing respective citations. I suggest adding context literature and maybe softening the statement as it is a topic of ongoing research. See for example [Abdar et. al. 2021] (https://www.sciencedirect.com/science/article/pii/S156625: who discuss the topic in detail also theoretically referring to an MLP with one hidden-layer such as yours.

  Thank you very much for the suggestion, we softened the claim and added references. The sentence reads now 'Modifying machine learning algorithms to output probabilistic forecasts is possible but requires either advanced techniques such as Bayesian computation or models ensembling. The link between numerical ensembles and probabilistic forecasts is an

active field of research (Collins et al., 2012; Rougier and Goldstein, 2014), thus in this exploratory study, we focus on classical ML algorithms, leaving probabilistic modeling for future work.'

- 2. Visualisation: While Figure 3 has undergone major improvements, unfortunately you have now dropped the PC labels for the individual plots. I suggest adding PC labels or adding A,B,C,D in the Figure and according assignment (A is PC1, etc.) in the caption. Otherwise, all visualisation adjustments are satisfactory

  Thank you for the spotting this omission, we have added the forgotten labels.

- 3. Citation of Kretschmer et.al 2017: Thank you for the clarification. I think that the statement of the authors is not completely true. The approach described in Kretschmer et al. discusses multiple testing and accounts for it as described in the SI (p2, Robustness of the causal precursor detection scheme): "Note that the causal discovery algorithm involves multiple hypotheses testing such that the significance parameter $\alpha$ should be considered as a hyper-parameter of the algorithm. As a conservative confidence estimate for a predictor (which is not really relevant for prediction), one can use the maximum p-value among all tested conditions in the PC algorithm. We also tested whether the obtained predictors are significant if we additionally control the false discovery rate (FDR) and found the lag-1 SPV index and v*T*100 over Eurasia (Fig. 3a) to be still significant (adjusted $p < 0.01$) but the other two region (Fig. 3a) slightly dropping in significance (adjusted $p \approx 0.15$)." Thus, I suggest softening the statement but do not see this as a requirement for the publication of this manuscript.

  Thank you very stressing all these details from their work. In their light, we agree that the sentence's claim should be lowered. The sentence now reads : 'their approach is efficient and provides convincing results but cannot scale to very large problems such as ours where we jointly analyze multiple levels and, being a two-step procedure, their methodology requires controlling for multiple testing, which, if not properly adjusted, is susceptible to selection bias in high-dimensional setups.'